# Comparison of Nucleosome Landscapes Between Porcine Embryonic Fibroblasts and GV Oocytes

**DOI:** 10.3390/ani14233392

**Published:** 2024-11-25

**Authors:** Minjun Zhao, Shunran Zhao, Zhaoqi Pang, Chunhui Jia, Chenyu Tao

**Affiliations:** College of Animal Science and Technology, Hebei Agricultural University, Baoding 071000, China; 15103177077@163.com (M.Z.); zhaoshunran9@163.com (S.Z.); 17731597059@163.com (Z.P.); jia18531453172@163.com (C.J.)

**Keywords:** nucleosome, GV oocytes, porcine embryonic fibroblast, transcriptional start site, gene activity

## Abstract

Nucleosomes serve as the basic structural components of chromatin and act as key regulators of gene expression and cellular function. In this study, we compared the nucleosome occupancy, distribution, and transcription of genes between two types of cells: fully grown germinal vesicle (GV) oocytes from big follicles (BF) and somatic cells (porcine embryonic fibroblast, PEF). The nucleosome occupancy in the BF promoter was significantly higher than in PEF. A reanalysis of published transcriptomes of fully grown GV oocytes and PEF identified 51 uniquely expressed genes in BF and 80 in PEF. The nucleosome distribution around gene transcription start sites (TSSs) correlated with expression levels in somatic cells, but results differed between BF and PEF. This research reveals the dynamic changes and crucial roles of nucleosome positioning and chromatin organization in different cell types, providing a theoretical basis for nuclear transplantation.

## 1. Introduction

Nucleosomes are the fundamental structural units of chromatin formed by DNA and histones. In eukaryotic cells, genomes are packaged into nucleosomes, constituting approximately 75–90% of the entire genome [1]. Nucleosomes are composed of roughly 147 base pairs of DNA that wrap around a histone octamer approximately 1.7 times; DNA linkers (20–54 bp) separate the nucleosomes. Nucleosomes act as essential structural units of genomes, establishing the foundation for the higher-order organization of chromatin [2,3]. A significant part of genomic DNA is constituted by nucleosomes, excluding specific functional areas like promoters and enhancers, which are relatively lacking in nucleosomes [4]. Recent studies suggest that the canonical nucleosome arrangement (−1, nucleosome depletion region [NDR], +1, +2) surrounding the transcription start site (TSS) is crucial for gene activation [5,6]. In our earlier research, we established a technique to profile the genome-wide nucleosome distribution map using only 1000 cells. By comparing the modified regions, we observed that developing oocytes exhibited a tendency for nucleosome loss and a more open chromosomal structure compared to fully mature oocytes [7].

Mammalian oocytes undergo a long and complex developmental process. As the volume of the oocyte increases, a mass of protein, mRNA, and energy substances, such as lipids, are reserved for use in fertilization and embryogenesis [8]. As with the development of single-cell sequencing (SCS), many new oocyte-specific transcription factors have been identified. They provide oocytes with a unique ability to force genome-wide epigenetic changes in somatic cells, which will, in turn, reprogram their nuclei into pluripotency; for example, the zinc finger protein GLIS1, a nonspecific transcription factor found to be enriched in unfertilized eggs and in embryos at the one-cell stage, which also plays an important role during zygotic genome activation (ZGA) and the preimplantation development of bovine embryos [9]. GLIS1 can replace c-myc with the KSOM formulation in human fibroblast. It directly interacts with Oct4, Sox2, and Klf4, independently of p53, facilitating various pathways that enhance reprogramming and the upregulation of additional transcription factors, such as NANOG, GATA4, FOXA2, and NKX2 [10]. The oocyte-specific transcription factor homeobox 6 (Obox6) has been reported to improve reprogramming in mouse cells [11]. Obox6 was initially recognized as a transcription factor uniquely associated with mouse oocytes [12]. It was then shown that ectopic expression of Obox6 occurred in mouse embryonic stem cells (mESCs) and early mouse preimplantation embryos, with the peak level of expression observed between the two-cell and morula stages [13,14]. Another possible explanation for ZGA is the chromatin structure. The genome of fully grown oocytes is silent, but exogenous DNA can be transcribed. Thus, it has been proposed that histones deposited by the mother serve as potential repressors [15,16].

Nucleosome occupancy and positioning in oocytes play an important role in reprogramming. Fully mature oocytes in the germinal vesicle (GV) stage later undergo meiosis, resulting in haploid cells that are prepared for fertilization, while the somatic cells undergo mitosis after DNA replication. Thus, we presume that considerable variation in chromosome structure and gene expression may exist between these two types of cells.

In our study, we used our previously published MNase-seq data of pig, fully grown GV oocytes (BF), porcine embryonic fibroblasts (PEF), and the currently published transcriptome data of BF and PEF. We compared the nucleosome distribution of the whole genome as well as those surrounding the TSSs between the two cell types. Specifically, expressed genes were identified in oocytes and PEFs. In PEFs, the nucleosomes surrounding the TSSs of differentially expressed genes were in accordance with the RNA-seq data; however, this was not the case in oocytes. These findings offer fresh perspectives on how nucleosome repositioning is regulated across different cell types, while also laying a foundation for further exploration of oocyte genetics.

## 2. Materials and Methods

### 2.1. Data Generation

Ovaries from 7 to 8-month-old commercial pigs were collected and transported to our lab in 1 h by a thermo with 37 °C normal saline. Then, the fully grown oocytes were selected with diameters of approximately 120 μm by a vacuum suction from the 5–8 mm big follicles (BF). While the porcine embryonic fibroblasts were preserved by our lab. All the MNase-seq libraries of 100 bp paired-end reads were sequenced using the Illumina HiSeq2000 system and were performed by Novogene (Beijing, China).

All pig MNase-seq and RNA-seq datasets utilized in this study were sourced from the sequence read archive (SRA) at national center for biotechnology information (NCBI) (http://www.ncbi.nlm.nih.gov/sra, accessed on 5 June 2024). The MNase-seq and RNA-seq data for porcine fully grown GV oocytes are accessible under the accession numbers, PRJNA347494 and SRX7575382, while those for PEF are cataloged as SRP090055 and SRX027095. The comprehensive datasets and associated experiments are detailed in Appendix A.

### 2.2. Calculation of Global Nucleosome Occupancy

The sequence reads were aligned to the Sus scrofa (pig) reference genome (Sscrofa 11.1, http://asia.ensembl.org/Sus_scrofa/Info/Index, accessed on 5 June 2024) by Bowtie2 (2.3.4.1), and only the uniquely aligned reads were kept for the analysis. To examine how the CpG sites (GC) content influences nucleosome occupancy, 10 kb genomic segments were categorized into five bins according to GC percentage, and nucleosome occupancy was quantified as fragments per kilobase per million (FPKM).

### 2.3. Genome-Wide Comparison of Nucleosome Occupancy

Sequence reads were aligned to the Sus scrofa (pig) reference genome Sscrofa 11.1 using Bowtie2, with only uniquely matched reads kept. Paired reads were treated as fragments, which were counted and normalized as FPKM to assess nucleosome occupancy.

To explore the relationship between GC content and nucleosome occupancy, 500 bp fragments were grouped into five categories based on GC content, and their FPKM values were calculated.

To compare the nucleosome occupancy with a genome-wide scan using 500 bp windows, FPKM values were computed for each window, and pairwise comparisons between samples were made to detect differences. Windows showing either a two-fold increase or a decrease in FPKM values were identified for further analysis.

### 2.4. Nucleosome Occupancy in Different Regions of the Genome

The nucleosome distribution across the genome was systematically evaluated by quantifying the proportion of nucleosome reads across distinct genomic regions, such as promoters, 5′ UTRs, exons, introns, 3′ UTRs, and intergenic areas. The sequence data for these regions aligned with the Sscrofa 11.1 reference genome and were sourced from the UCSC Genome Bioinformatics database. Using BEDTools (version 2.16.2) [17], nucleosome occupancy levels were computed for each genomic region, and occupancy ratios were derived. These ratios were then compared against the overall distribution across the genome, providing deeper insights into nucleosome localization patterns in different regions.

### 2.5. Nucleosome Distribution Profiles

The gene annotation data were sourced from the UCSC Genome Database. Nucleosomes located within 1 kb upstream and downstream of each transcription start site (TSS) were gathered. This 2 kb region was divided into 10 bp bins, and the fragments per kilobase per million (FPKM) values or the nucleosome fragment counts in each bin were computed to generate a detailed nucleosome distribution profile around the TSS [17].

### 2.6. Analysis of Gene Expression

RNA-seq data for both oocytes and porcine embryonic fibroblasts (PEF) were retrieved from the NCBI database. The sequence reads were mapped to the UCSC genes (Sscrofa 11.1 reference version) using the HISAT-3N tool, and only uniquely aligned reads were used for the subsequent analysis. HTSeq was utilized to quantify the read counts per gene, and these were normalized to FPKM (fragments per kilobase per million) values. In this study, genes were categorized into three groups based on their expression levels: the top 5% most highly expressed genes, silent genes (genes which were not in the transcriptome data, that is, genes that were not expressed), and the other genes (genes which were expressed except those top 5%) [7,18]. Nucleosome distribution patterns around the transcription start sites (TSS) were then generated following the previously outlined methods.

### 2.7. Statistical Analysis

The obtained data were statistically analyzed using SPSS version 21.0. The normal distribution and homogeneity of variance were tested. Student’s independent t test was employed to reveal the significance between BF and PEF groups. *p*-values < 0.05 and <0.01 were considered significant differences and extremely significant differences, respectively.

## 3. Results

### 3.1. Global Nucleosome Organization and Occupancy of Fully Grown GV Oocytes and PEF

To characterize the differences in nucleosome organization and occupancy between the two cell types, the MNase- and RNA-seq data of BF and PEF were collected and analyzed comprehensively. The basic quality and analysis information of the data are shown in Appendix A. About 72% of the genome in fully grown GV oocytes was occupied by nucleosomes, while 76% of the genome in PEFs was occupied by nucleosomes, suggesting that the majority of genomic DNA is organized into nucleosome structures.

To provide more clarity on the differences in nucleosome distribution and occupancy, the genome was divided into different functional regions: promoter, 5′ UTR, 3′ UTR, 1st exon, 1st intron, other exons, other introns, and downstream and intergenic regions. The nucleosome occupancy data for each region are shown in Figure 1A and Appendix A. The results indicate that the nucleosome occupancy in genic regions was 61.4% for BF and 55.42% for PEF, which is far greater than the 26.56% classified as genic regions in the genome categorized. Nevertheless, the nucleosome occupancy in the intergenic regions (38.6% and 44.58% for BF and PEF, respectively) was lower than that in the genome (73.44%). Nucleosome occupancy in the BF promoter reached 4.85%, significantly surpassing the 3.3% observed in PEF (*p* < 0.05). A comparison of nucleosome occupancy ratios across regions of BF and PEF highlights distinct patterns of nucleosome distribution between the two cell types. The nucleosome occupancy in genic regions was notably higher in BF compared with PEF, that is, 5′ UTR (0.83% vs. 0.57%, *p* < 0.05), 3′ UTR (2.69% vs. 1.89%, *p* < 0.05), exons (5.95% vs. 3.78%, *p* < 0.05), and introns (47.08% vs. 45.87%, *p* < 0.05). These changes may cause different transcriptional situations and functions of the two types of cells.

An intriguing parabolic relationship emerged between the nucleosome occupancy and GC content in both samples, where nucleosome occupancy surged with rising GC content before ultimately tapering off. (Figure 1B).

### 3.2. Different Nucleosome Occupancy Between the Two Cell Types

To characterize the diverse chromosomal states between the two cell types, we conducted a pairwise comparison of genome-wide nucleosome occupancy between the samples (Figure 2A), using a 10 kb window to scan the genome. The results demonstrate changes in nucleosome occupancy across all BF and PEF chromosomes. Overall, nucleosome occupancy was deficient when comparing PEF with BF, which may have a great impact on global transcription. A more detailed analysis was performed by scanning the entire genome with a 500 bp window and identifying windows with a two-fold change in nucleosome occupancy. In total, 13,674 windows displayed more than a two-fold increase in nucleosome occupancy in BF compared to PEF, while 4374 windows in BF exhibited decreased occupancy (Figure 2B). This analysis confirmed that nucleosome occupancy was reduced in PEF, indicating a more open chromatin structure compared to BF.

Since the nucleosome distribution around the TSS is critical in regulating gene expression, we next explored the nucleosome distribution within 1 kb of the TSS flanking region. The nucleosome distribution showed a noticeable increase surrounding TSSs in BF (Figure 2C,E), which was in accordance with the typical nucleosome arrangements of silent genes; however, it was quite different from that in PEF, which showed little nucleosome occupancy depletion at TSSs and noticeable +1 and +2 nucleosomes (Figure 2D,E). This suggested that the variation of nucleosome arrangements surrounding TSSs may contribute to different cell functions between the two cell types. In different cells, the precise position of nucleosomes will vary more or less (deviation), centered on a most preferred position. This deviation of nucleosome position in each unit of the cell is known as fuzziness. Nucleosome occupancy shifts are also affected by nucleosome fuzziness, which influences the precision of nucleosome positioning; thus, we calculated the reads moving from the nucleosome dyad to the neighboring linker regions. The results showed decreased nucleosome occupancy but increased fuzziness in PEFs (Figure 3A,B).

### 3.3. Specific Gene Expression in Two Cell Types

Nucleosome occupancy and distribution are the major factors affecting gene expression. Next, we reanalyzed the currently published transcriptome of GV oocytes and the PEF of porcine oocytes. Pearson correlations between the samples are shown in Figure 4A. The intraclass correlation coefficients were all greater than 0.9, while they were significantly higher than those between groups, indicating the inconsistency in different types of cells. From the RNA-seq data, a total of 51 genes in BF and 80 genes in PEF were identified as uniquely expressed (Appendix A). A GO analysis of the 51 and 80 genes revealed that these genes were primarily associated with biological processes, such as cell function, cellular interaction, and binding. However, a total of 24 genes uniquely expressed in fully grown GV oocytes took part in the process of reproduction and reproductive processes (Figure 4B). Over 30 genes were involved in cellular component organization and developmental processes in PEF (Figure 4C). Five genes from BF and PEF were selected for the validation of RNA-seq results using q-PCR. The expression levels of ZP3, ZP4, OOEP, SYCN, and DAPP5 in BF, and THBS1, ACTA2, SPARC, FN1, and COL1A1 in PEF, showed consistency with the RNA-seq data. (Figure 4D,E).

### 3.4. Nucleosome Configurations Around TSSs Correlate with Gene Activity

Considering the essential role of the canonical nucleosome arrangement (1, NDR, +1, +2) around TSSs in gene expression, we analyzed the nucleosome organization surrounding the TSS of all genes in both BF and PEF (Figure 2C,D). As expected, the nucleosome arrangements of 1, NDR, +1, and +2 around the TSSs in PEF and BF show significant differences. To further explore the relationship between nucleosome distribution and gene expression, we classified 22,861 porcine genes into three groups based on PEF RNA-seq data: the top 5% highly expressed genes, silent genes, and all other genes (see Section 2). In PEF, we observed the canonical nucleosome arrangement of NDR, +1, and +2 nucleosomes around the TSSs of both the top 5% expressed genes, while an increase in nucleosome occupancy surrounding TSSs was found in silent genes (Figure 5B), which was in accordance with the typical nucleosome arrangement of highly expressed genes and silent genes. However, the results for BF were not consistent with those in PEF. The genes were divided into three groups based on the RNA-seq data of BF. Unexpectedly, the results of the nucleosome distribution of the three types of genes all showed a silent type with an increased nucleosome occupancy at TSSs, which was not consistent with the RNA-seq results (Figure 5A).

## 4. Discussion

Nucleosomes make up most of the genome and are thought to be displaced by transcription factors in regions that direct gene expression. The most effective strategy for constructing comprehensive genome-wide nucleosome maps involves the digestion of chromatin with micrococcal nuclease to produce mononucleosomes, which are then analyzed through sequencing (MNase-seq), allowing for the precise mapping of nucleosome positions across the genome. Nucleosome occupancy and distribution in the nucleus are a new epigenetic method for elucidating biological changes. Thus, we analyzed the nucleosome distribution and occupancy difference to compare the two cell types.

It is well known that the oocyte is the largest cell in the mammals and contains a large amount of cytoplasmic inclusion, which is an important factor in the subsequent reprogramming and will be the energy source in the development of embryos [19]. According to the results of our analysis of the RNA-seq data, the uniquely highly expressed genes in BF, such as ZP2, ZP3, and ZP4, are all important proteins in the composition of the zona pellucida. DAPP5 is uniquely expressed in fully grown GV oocytes, as documented in previous studies; it is the key factor in the reprogramming and development of embryos. Porcine embryonic fibroblast (PEF) is a type of somatic cell which is widely used in somatic cell nuclear transplantation (SCNT) [20,21,22]. SCNT is used to replace the nucleus of the oocyte with PEF. The further study of downstream targets of DAPP5 could be selected by the nucleosome sequencing data of PEF. An analysis like this would reveal how regulatory factors in the cytoplasm of oocytes interact with somatic cells to start reprogramming. Reprogramming encompasses a sequence of changes that unfold in a particular order and within a set timeframe [23]. Nucleosome remodeling represents a dynamic process that occurs throughout reprogramming. Our prior research indicated that donor PEF chromatin, especially the X chromosome, exhibited increased openness 10 h post-transfer into porcine oocytes, with a corresponding decrease in nucleosome occupancy at the promoters [24]. Thus, comparing the transcriptome of oocytes and the nucleosome distribution and occupancy of PEF is crucial for understanding the mechanism of reprogramming.

Fertilization is considered one of nature’s greatest feats, starting with a spermatocyte merging with an oocyte, transforming these two terminally differentiated germ cells into a totipotent zygote. Major transformations take place to facilitate a sequence of critical biological events, including oocyte activation, the maternal-to-zygote transition aligned with ZGA, and the subsequent initial cell-fate decision along with lineage-specific differentiation. SCNT is an artificial method of producing embryos, during which reprogramming, including DNA demethylation and histone acetylation, occurs at the onset of the activation of cloned embryos. Up until 4–8 cells, the genome begins to translate globally. There are two views regarding the molecular pathways that activate ZGA. One is that transcription is suppressed due to high histone content in the chromatin of the zygote, while transcription is gradually restored as histone content in each cell decreases with the initiation of cleavage [25]. The other is that key maternal transcription factors activate the expression of key genes, leading to the activation of zygotic genomes [23]. Since the study of nucleosome positioning and occupancy is the primary structure of chromatin, it can reflect the histone content and arrangement and investigate the expression of some key genes. This is a novel approach to improving the efficiency of SCNT.

The change in nucleosome distribution and occupancy in different functional elements of the genome can alter the expression of genes to a large extent and even the state of the cell, especially in the promoter region. In this study, we found that the nucleosome occupancy in PEFs was 1.3%, while it was 2.85% in BF (Figure 1A,B), which indicated that transcription in PEF may be more active than that in BF. The GC content correlates with gene density, meaning that the GC content in gene regions is higher than that in non-gene regions. Promoters mostly contain CpG islands and are always of the highest GC content in the gene body. The 10 kb fragments with low GC content in this study were mostly in the non-gene region, while the high GC contents, even promotors, were in the gene region. Thus, a parabolic relationship was observed between nucleosome occupancy and GC content between the two samples, as with the increase in GC content, nucleosome occupancy increased, and then declined. We know that CpG islands can regulate gene expression and silencing through DNA methylation and non-methylation. On the other hand, the results of this experiment indicate that promoters can also regulate the number and distribution of nucleosomes through high GC content, thus regulating gene expression.

Nucleosome fuzziness means “standard deviation of tag distances from a consensus nucleosome midpoint”. Nucleosome fuzziness that increases toward the 3′ ends of genes is thought to reflect the translational fluidity of nucleosomes on DNA, while higher-order chromatin compaction is anticipated to limit this fluidity. However, in the previous study, such a reduction in fuzziness was not observed during a meiotic developmental process [26].

Furthermore, the nucleosome distribution surrounding the TSSs in BF showed a typical silent gene (Figure 2C), while PEF showed a typical active gene. Although prior research suggests that the loss of nucleosomes in promoters can lead to elevated transcriptional activity, BF shows significantly low global transcriptional activity [5]. Therefore, it can be concluded that gene transcriptional activity is influenced by both nucleosome occupancy and the spatial arrangement of nucleosomes within the promoter region. Generally, the canonical nucleosome organization includes NDR, as well as +1 and +2 nucleosomes surrounding the TSS region, which is consistent with the expression quantity of genes. However, interestingly, we found that the results in the BF did not agree with that. Surprisingly, the nucleosome distribution among the three gene types (the top 5% highly expressed genes, expressed genes, and silent genes) revealed a silent type characterized by increased nucleosome occupancy around the TSSs, which was not consistent with the RNA-seq results (Figure 5B). This may be because, during the growth of oocytes, many mRNAs, proteins, lipids, and many other energetic substances and regulation factors accumulate in the cytoplasm, which results in an increase in oocyte volume. Actively growing oocytes display high transcriptional activity, while fully mature oocytes show reduced transcriptional activity [27]. So, the RNA-seq data could reflect the RNAs in the cytoplasm, but not the gene expression. Thus, the nucleosome distribution surrounding the TSSs and the RNA-seq results could not be reproduced. According to our results, dynamic nucleosome positioning is related to gene expression; however, it is not applicable to certain cell types, such as mammalian oocytes and embryos.

## 5. Conclusions

Our results revealed the landscapes of porcine embryonic fibroblasts and GV oocytes. The nucleosome occupancy and arrangement in the BF promoter were silent. Nucleosome distribution surrounding the TSSs of genes was consistent with the expression levels in somatic cells; however, the results in BF did not agree with those in PEF.

## Figures and Tables

**Figure 1 animals-14-03392-f001:**
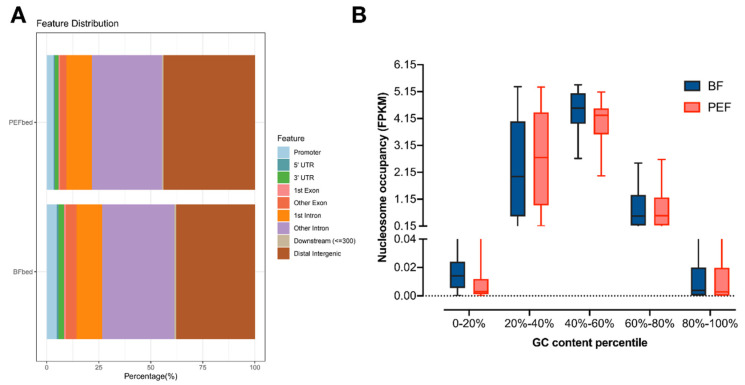
Global nucleosome occupancy of PEFs and BFs. (**A**) Bar plot showing the comparison ratio of the functional element distribution between PEFs and BFs. (**B**) A box plot illustrating the correlation between nucleosome occupancy and GC content. The blue box represents BFs, while the red box corresponds to PEFs.

**Figure 2 animals-14-03392-f002:**
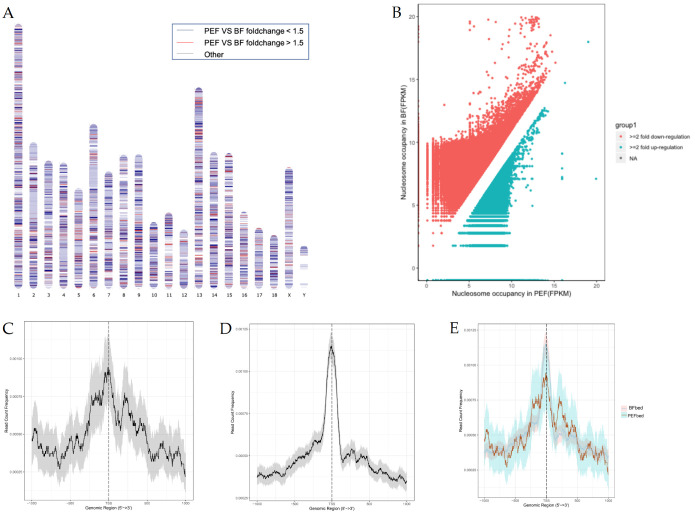
Nucleosome occupancy comparison between PEFs and BFs. (**A**) A genome-wide comparison of nucleosome occupancy between PEFs and BFs was performed. The color scheme represents the shifts in nucleosome occupancy for every 10 kb region between the two samples: bright red highlights a significant ≥1.5-fold increase in PEF nucleosome occupancy, while deep green marks a pronounced ≤1.5-fold decrease; gray indicates no changes in nucleosome occupancy; and white indicates regions with no enrichment. (**B**) A differential analysis of nucleosome occupancy between PEFs and BFs was conducted. Red highlights a dramatic ≥2-fold increase in nucleosome occupancy in BF, whereas blue reveals a notable ≤2-fold decrease in nucleosome occupancy in SF. (**C**–**E**) Average profile of MNase peaks surrounding the TSS region. Comparison of MNase peak distributions between BFs (**C**), PEFs (**D**), and merge (**E**).

**Figure 3 animals-14-03392-f003:**
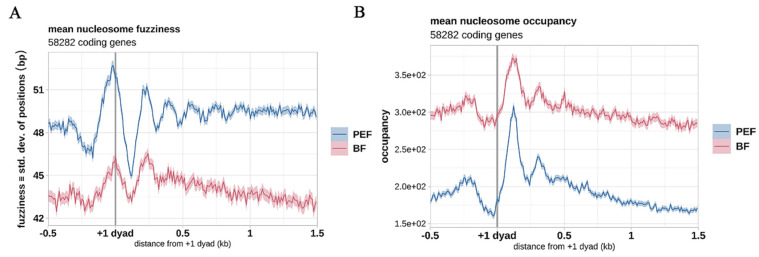
Average profile plot showing nucleosome occupancy in regions containing dynamic nucleosomes. (**A**,**B**) Nucleosomes showing an increase in PEF fuzziness (**A**) and an occupancy decrease (**B**).

**Figure 4 animals-14-03392-f004:**
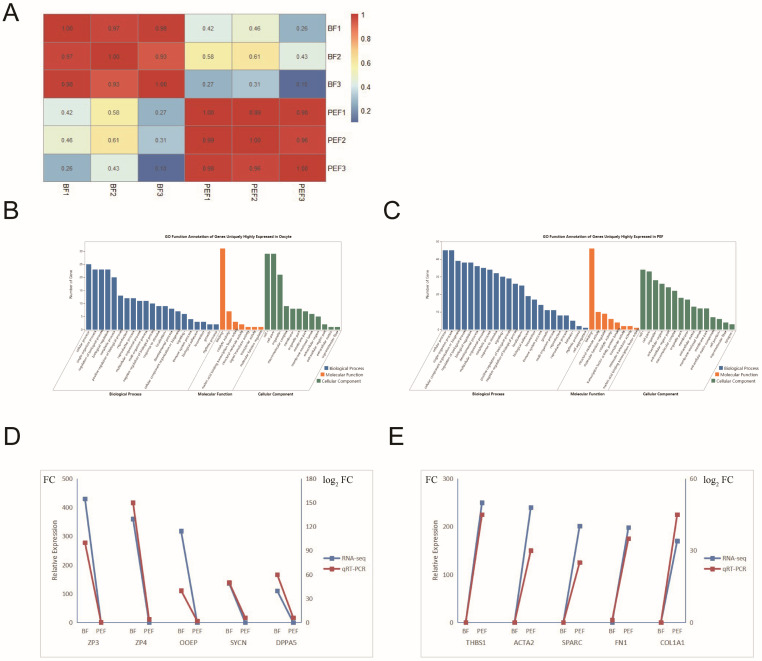
The RNA-seq analysis of BFs and PEFs. (**A**) Pearson correlations among the replicate samples of BFs and PEFs. (**B**,**C**) The GO functional annotation of uniquely expressed genes in BFs (B) and PEFs (**C**). The *x*-axis shows the three categories and the detailed GO terms, while the *y*-axis shows the number of genes. (**D**,**E**) The q-PCR validation of the RNA-seq data. ZP3, ZP4, OOEP, SYCN, and DAPP5 in BFs (**D**) and THBS1, ACTA2, SPARC, FN1, and COL1A1 in PEFs (FC, fold change) (**E**).

**Figure 5 animals-14-03392-f005:**
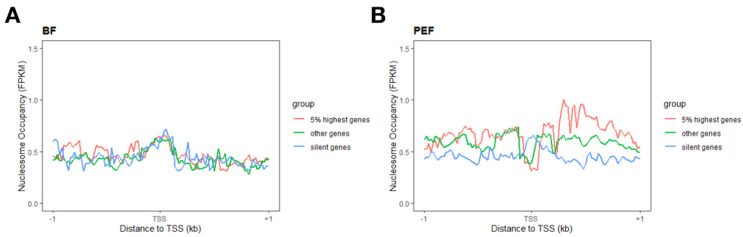
The patterns of nucleosome occupancy around TSSs reveal striking differences among the top 5% most expressed genes (red), silent genes (green), and the remaining genes (blue) in both BFs (**A**) and PEFs (**B**).

## Data Availability

The accession numbers of the MNase- and RNA-seq data of porcine fully grown GV oocytes were PRJNA347494 and SRX7575382, while the accession numbers of the MNase- and RNA-seq data of PEF were SRP090055 and SRX027095.

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
