# Peer review of "Comparison of Nucleosome Landscapes Between Porcine Embryonic Fibroblasts and GV Oocytes"

_animals, 2024, doi:10.3390/ani14233392_

Round 1
Reviewer 1 Report
Comments and Suggestions for Authors
In this manuscript, the authors profile nucleosome occupancy by MNase assays in porcine germinal vesicle stage oocytes (BF) and embryonic fibroblasts (PEF). They found that the nucleosome occupancy in the BF promoter was significantly higher than in PEF. Then they performed a combined analysis of Nase-seq data and previously published RNA-seq data and found that nucleosome occupancy does not correlate with gene expression in BFs. These results provide new insights into the regula-81 tion of nucleosome repositioning between different cell types. However, there are a few areas where the article could be improved.
Line86-91 Briefly introduce the progress of data generation, including cell isolation, sequencing platform and so on.
Line99 Sus scrofa (pig) reference genome Sscrofa 11.2? Are you sure?
Line101-102 Polish this sentence.
Line109 What does “SF” mean? Provide its full name.
Line117 Revise “50 UTR” as “5′ UTR”.
Line118 Revise “30 UTR” as “3′ UTR”.
Line116-118 These regions maybe overlapped causing by AS, overlapping genes and other factors. Please tell me how to deal this case.
Line119 Why use the specific sequence information corresponding to the reference genome Sscrofa 10.2? The sequence reads were mapped to the Sus scrofa (pig) reference genome (Sscrofa 11.1, http://asia.ensembl.org/Sus_scrofa/Info/Index). So, I think the specific sequence information corresponding to Sscrofa 11.1, rather than Sscrofa 10.2 should be used.
Line131 susScr3 is Sscrofa 10.2 in ensemble. Sscrofa 11.1 is a better reference genome and has been released a few years. So, Sscrofa 11.1 should be used in this study in my view.
Line135 I don’t know how to define “silence genes” and “other genes”. Please provide the criteria for detail.
Line141, 162-164 They are methods and had been provide in method section. So I suggest removing them
Line174-184 1.5-fold change and 2-fold change were both used in differential analysis. Why?
Line209 It would be helpful to explain whatnucleosome “fuzziness” mean and what is the biological relevance.
Line213-214 I cannot well understand this sentence. Does the authors mean that the intraclass correlation coefficients are significantly lower than that between groups? However, as shown in Fig 4A, the intragroup correlation coefficients are higher than intergroup correlation coefficients.
Line217 Polish this sentence.
Fig 4D and 4E What does the right Y mean.
Author Response
Reviewer 2
- Line86-91 Briefly introduce the progress of data generation, including cell isolation, sequencing platform and so on.
Response: Thank you for your suggestions. The specific information of the cell isolation, sequencing platform were added at line 97-102 in the revised version.
“Ovaries from 7-8-month-old commercial pigs were collected and transported to our lab in 1 h by a thermo with 37˚C normal saline. Then, the fully grown oocytes were selected with diameters of approximately 120 μm by vacuum suction. While the porcine embryonic fibroblast were preserved by our lab. All the MNase-seq libraries of 100-bp paired-end reads were sequenced using the Illumina HiSeq2000 system and
were performed by Novogene (Beijing, China).”
- Line99 Sus scrofa (pig) reference genome Sscrofa 11.2? Are you sure?
Response: Thank you for your question and we are really sorry about the mistake we made. We did the analysis with Sscrofa 11.1 version (which were highlighted at line 111), however we made a mistake here, and we have changed Sscrofa 11.2 to Sscrofa 11.1. Sorry again for our mistake, we will pay attention next time.
- Line101-102 Polish this sentence.
Response: Thank you for your suggestion, we have changed the sentence into “Paired reads were treated as fragments, which were counted and normalized as FPKM to assess nucleosome occupancy. ” at line 118-120 in the revised version.
- Line109 What does “SF” mean? Provide its full name.
Response: Thank you for your question and we are sorry for our unclear statement. We have changed it into “To compare the nucleosome occupancy, with a genome-wide scan using 500 bp windows. FPKM values were computed for each window, and pairwise comparisons between samples were made to detect differences. Windows showing either a two-fold increase or decrease in FPKM values were identified for further analysis.” at line 124-127 in the revised version.
- Line117 Revise “50 UTR” as “5′ UTR”.
Response: Thank you for your comments and suggestion for improving our manuscript. We are sorry for the mistake, and we have revised 50 UTR to 5′ UTR at line 131.
- Line118 Revise “30 UTR” as “3′ UTR”.
Response: Thank you for your comments and suggestion for improving our manuscript. We are sorry for the mistake, and we have revised 30 UTR to 3’ UTR at line 131.
- Line116-118 These regions maybe overlapped causing by AS, overlapping genes and other factors. Please tell me how to deal this case.
Response: Thank you for your comments and suggestion for improving our manuscript. We also consider that this may be the case, however we think that this analysis is more focused on the functional elements regions of each gene and does not take these factors into account.
- Line119 Why use the specific sequence information corresponding to the reference genome
Response: Thank you for your comments and suggestion for improving our manuscript. We are sorry for the mistake. Because there are some gene analyses in our subsequent research that are not included in this paper, and some specific sequences are used in these analyses. I am sorry for that and have deleted it in the revised version.
- Sscrofa 10.2? The sequence reads were mapped to the Sus scrofa (pig) reference genome (Sscrofa 11.1, http://asia.ensembl.org/Sus_scrofa/Info/Index). So, I think the specific sequence information corresponding to Sscrofa 11.1, rather than Sscrofa 10.2 should be used.
Response: Thank you for your comments and suggestion for improving our manuscript. We did the analysis with Sscrofa 11.1 version (which were highlighted at line 111), however we made a mistake here, and we have corrected it. Sorry again for our mistake, we will pay attention next time.
- Line131 susScr3 is Sscrofa 10.2 in ensemble. Sscrofa 11.1 is a better reference genome and has been released a few years. So, Sscrofa 11.1 should be used in this study in my view.
Response: Thank you for your comments and suggestion for improving our manuscript. We did the analysis with Sscrofa 11.1 version (which were highlighted at line 111), however we made a mistake here, and we have corrected it. Sorry again for our mistake, we will pay attention next time.
- Line135 I don’t know how to define “silence genes” and “other genes”. Please provide the criteria for detail.
Response: Thank you for your question and suggestions. In this study, genes were categorized into three groups based on their expression levels: the top 5% most highly expressed genes, silent genes (genes which were not in the transcriptome data, that is, genes that were not expressed), and the other genes (genes which were expressed except those top 5% ).
Thank you for your suggestion, and we have added this information at line 151-154 in the revised version.
- Line141, 162-164 They are methods and had been provide in method section. So I suggest removing them
Response: Thank you for your suggestion, and we have deleted these sentences in the revised version.
- Line174-184 1.5-fold change and 2-fold change were both used in differential analysis. Why?
Response: Thank you for your question. We used different value of fold-change here for different purpose. First, to characterize the diverse chromosomal states between the two cell types, we compared the genome-wide nucleosome occupancy of the samples in a pairwise manner. 1.5-fold change was used here for the approximate descriptions of change between the two cell types. However, further study of a two-fold change was conducted to explore the specific differences to select some significant genes.
- Line209 It would be helpful to explain what nucleosome “fuzziness” mean and what is the biological relevance.
Response: Thank you for your question. “In different cells, the precise position of nucleosomes will vary more or less (deviation), centered on a most preferredposition. This deviation of nucleosome position in each unit of the cell is known as fuzziness. Nucleosome occupancy shifts are also affected by nucleosome fuzziness, which influences the precision of nucleosome positioning; thus, we calculated the reads moving from the nucleosome dyad to the neighboring linker regions. ” This information was added in the revised version.
- Line213-214 I cannot well understand this sentence. Does the authors mean that the intraclass correlation coefficients are significantly lower than that between groups? However, as shown in Fig 4A, the intragroup correlation coefficients are higher than
intergroup correlation coefficients.
Response: Thank you for your comments we are sorry for the mistake. Here we meant that “the intraclass correlation coefficients were significantly higher than that between groups”. We have revised it in the revised version.
- Line217 Polish this sentence.
Response: Thank you for your suggestion. We have revised this sentence into “GO analysis of the 51 and 80 genes revealed that these genes were primarily associated with biological processes such as cell function, cellular interaction, and binding.” in the revised version.
- Fig 4D and 4E What does the right Y mean.
Response: Thank you for your question and suggestion. We are sorry for the unclear description in Figure 4Dand 4E. Y meant log2 fold change on left and fold change on right. We have corrected it in the revised version.

Reviewer 2 Report
Comments and Suggestions for Authors
I have completed the review of your manuscript, and my comments are provided below. The authors compared the nucleosome landscapes between porcine GV stage oocytes and porcine embryonic fibroblasts. The most significant issue is the lack of specific cell experiments throughout the article. Additionally, several important questions raised by the reviewer need to be carefully addressed.
You have provided the background, methods, results, and conclusion, but the purpose of the study is missing in the abstract section.
Why did you only select big follicles? How did you define big, medium and small follicles?
Does BF (big follicles) equate to oocytes? In your study, two types of cells were evaluated: GV stage oocytes and porcine embryonic fibroblasts, not BF.
How did you determine statistical significance? What statistical analysis method was used in your study?
The authors need to carefully revise the manuscript again.
Author Response
1.You have provided the background, methods, results, and conclusion, but the purpose of the study is missing in the abstract section.
Response: Thank you for your comments and suggestion for improving our manuscript.
We have added the purpose to the abstract. We added " To furnish theoretical insights and data that support the process of cell reprogramming subsequent to nuclear transplantation." in line 21-22 of the manuscript.
- Why did you only select big follicles? How did you define big, medium and small follicles?
Response: Thank you for your comments and suggestion for improving our manuscript.
In our previous research, the growing oocytes (which are in the stage of active transcription and accumulation of mRNA and proteins) were collected with the diameter of approximately 20–30 μm from the small follicles (SF) which were less than 50 μm, while the fully grown oocytes (which have completed the stage of gene expression and accumulation, and have reached their final size) were collected from the 5–8 mm big follicles (BF) [1].
- Tao C, Li J, Chen B, Chi D, Zeng Y, Liu H. Genome-scale identification of nucleosome organization by using 1000 porcine oocytes at different developmental stages. PLoS One. 2017 Mar 23;12(3):e0174225. doi: 10.1371/journal.pone.0174225. PMID: 28333987; PMCID: PMC5363847.
We have added this information at line 87-89 in the revised version.
- Does BF (big follicles) equate to oocytes? In your study, two types of cells were evaluated: GV stage oocytes and porcine embryonic fibroblasts, not BF.
Response: Thank you for your comments on the manuscript. BF (big follicles) in the manuscript meant the fully grown oocytes, the name of BF referred to germinal vesicle (GV) oocytes derived from large follicles. Since this MNase-seq data were derived from our previous research, the naming continues from the previous article.
- How did you determine statistical significance? What statistical analysis method was used in your study?
Response: Thank you for your question and suggestion on the manuscript. The obtained data were statistically analyzed by SPSS version 21.0. The normal distribution and homogeneity of variance were tested. The Student’s independent t test was employed to reveal the significance between BF and PEF groups. P-values <0.05 and <0.01 were considered significant differences and extremely significant differences, respectively. We have added these information in “Statistical Analysis” at line 157-162 in the revised version.

Round 2
Reviewer 1 Report
Comments and Suggestions for Authors
I am happy to note that authors have worked hard to improve the manuscript and all the comments raised by me were not only answered but also implemented to improve the manuscript. Here is a minor comment.
1、For the classification criteria for "silence genes" and "other genes”, as well as the definition of nucleosome "fuzziness", additional supporting information or references could enhance the scientific rigor and reliability.
2、In Figures 4D and 4E, it is recommended to clearly describe the meanings of the left and right Y-axes in the figure legend.
3、In this manuscript, the authors corrected the reference genome version. However, I don't know why the references with same version were downloaded from two databases, Ensembl and UCSC. Are there any difference between them?
Author Response
Reviewer 1
1、For the classification criteria for "silence genes" and "other genes”, as well as the definition of nucleosome "fuzziness", additional supporting information or references could enhance the scientific rigor and reliability.
Response: Thank you for your comments and suggestion for improving our manuscript. The method for the classification of genes with different expression is based on the reference[1’2]. In their studies, RNA-seq data are also classified according to the expression of genes. We have quoted this article in the revised version at line 154. The information for “fuzziness” was cited from (Zhang et al., 2011) at line 356 [3].
- Huang K, Zhang X, Shi J, et al. Dynamically reorganized chromatin is the key for the reprogramming of somatic cells to pluripotent cells. Sci. Rep. 2015;5, 17691.
- Tao C, Li J, Zhang X, et al. Dynamic Reorganization of Nucleosome Positioning in Somatic Cells after Transfer into Porcine Enucleated Oocytes. Stem Cell Reports. 2017;9(2):642-653.
- Zhang L, Ma H, Pugh BF. Stable and dynamic nucleosome states during a meiotic developmental process. Genome Res. 2011;21(6):875-884.
2、In Figures 4D and 4E, it is recommended to clearly describe the meanings of the left and right Y-axes in the figure legend.
Response: Thank you for your question and suggestions. We have incorporated pertinent information into the result picture, with the left-hand coordinates representing log2FC and the right-hand coordinates indicating fold change (FC). Supplementary details have been appended at line 224 in the revised manuscript.
3、In this manuscript, the authors corrected the reference genome version. However, I don't know why the references with same version were downloaded from two databases, Ensembl and UCSC. Are there any difference between them?
Response: Thank you for your question. The pig reference genomes obtained from both the Ensembl and UCSC databases are identical, having been released in 2017. They have the same total gene length and identical sequencing lengths for each chromosome.
Figure 1 Comparison of Reference Genomes between Ensembl and UCSC. Panels A-B depict the UCSC reference genome and the assembly results for each chromosome, while Panels C-D show the Ensembl reference genome and the assembly results for each chromosome.

Reviewer 2 Report
Comments and Suggestions for Authors
This revised manuscript has been improved. I have no further comments.
Author Response
Thank you for your suggestion and comments for our manuscript.